# Sparse Low-rank Adaptation of Pre-trained Language Models

**Ning Ding**[1*], **Xingtai Lv**[1*], **Qiaosen Wang**[4], **Yulin Chen**[2]
**Bowen Zhou**[1†], **Zhiyuan Liu**[2,3†], **Maosong Sun**[2,3†]

[1]Department of Electronic Engineering, Tsinghua University
[2]Department of Computer Science and Technology, Tsinghua University
[3] BNRIST, IAI, Tsinghua University, [4]Department of Statistics, The University of Chicago
dn97@mail.tsinghua.edu.cn, lvxt20@mails.tsinghua.edu.cn

## Abstract

Fine-tuning pre-trained large language models in a parameter-efficient manner is widely studied for its effectiveness and efficiency. The popular method of low-rank adaptation (LoRA) offers a notable approach, hypothesizing that the adaptation process is intrinsically low-dimensional. Although LoRA has demonstrated commendable performance, it is implemented with a fixed and unalterable intrinsic rank that might not always be the ideal choice. Recognizing the need for more flexible adaptation, we extend the methodology of LoRA to an innovative approach we call sparse low-rank adaptation (SoRA) that enables dynamic adjustments to the intrinsic rank during the adaptation process. We achieve this through the incorporation of a gate unit optimized with proximal gradient method in the training stage, controlling the cardinality of rank under the sparsity of the gate. In the subsequent inference stage, we eliminate the parameter blocks corresponding to the zeroed-out ranks, to reduce each SoRA module back to a concise yet rank-optimal LoRA. Our approach strengthens the representation power of LoRA by initializing it with a higher rank, while efficiently taming a temporarily increased number of parameters via updating in a sparse way. We further introduce a sparsifying scheduler for SoRA, aiming to examine the impact of the number of non-zero parameters on the model's memorization and generalization. Our experimental results demonstrate that SoRA can outperform other baselines even with 70% retained parameters and 70% training time.

## 1 Introduction

Adapting large-scale pre-trained language models (Devlin et al., 2019; Brown et al., 2020; He et al., 2020; Bommasani et al., 2021; Han et al., 2021; Touvron et al., 2023) in a parameter-efficient (He et al., 2022; Ding et al., 2023; Hu et al., 2023) manner is increasingly gaining traction within the research community. The methods of this paradigm typically keep most of the parameters of the underlying model unchanged, either insert additional trainable parameters into the model (Houlsby et al., 2019; Li and Liang, 2021), or specify a small number of parameters (Zaken et al., 2021; Liu et al., 2021; Su et al., 2023) to be trainable or reparameterize the adaptation process into a more efficient form (Hu et al., 2021; Qin et al., 2021). They have been validated to be effective across various models and tasks, often yielding comparable or even better results than full-parameter fine-tuning.

The development potential of parameter-efficient fine-tuning became evident after extensive validation of its performance. These methods offer the opportunity to adapt the base model to fit any data, allowing for enhancements and customization of language models tailored to specific tasks and personalized user characteristics. Due to the lightweight nature of the optimized parameters, they can be seamlessly plugged into the model, allowing targeted enhancements to be made. Among these methods, low-rank adaptation (LoRA (Hu et al., 2021)) is considered one of the most efficient methods at present. It assumes that the change of the model's parameters after adaptation is "intrinsically low-dimensional" and performs adaptation by optimizing the matrix obtained from low-rank decomposition. LoRA avoids forward propagation latency caused by inserting additional neural modules while demonstrating stable performance. Although effective, the setup of the intrinsic rank (normally as a hyperparameter) is still unclear. Intuitively, a larger rank brings larger optimization space and creates the capacity to handle more challenging tasks. However, in practice, the optimal intrinsic rank would vary according to multiple factors such as the backbone model and the task.

Given the enormous computational cost of

---
[*] equal contributions
[†] corresponding authors

searching hyperparameters on large-scale models (such as GPT-3 (Brown et al., 2020) with 175 billion parameters and LLaMA (Touvron et al., 2023) with 700 million to 65 billion parameters), developing a method based on adaptive ranks is a natural approach. Some existing work has attempted to explore this direction (Valipour et al., 2022; Zhang et al., 2023), but they are largely heuristic or introduce additional costs. In this paper, we propose SoRA, a simple, effective, and automated method for adaptive parameter-efficient fine-tuning. We introduce a gating module with a proximal gradient decent update under L1 regularization to control the sparsity of the updated matrices. After training, the zero entry of the gating vector records the columns of the down-projection matrix and the rows of the up-projection matrix, which can be simply dropped and stored in a more parameter-efficient manner. Compared to other adaptive approaches, the proximal gradient method has a clear mathematical meaning and does not have to involve other computations and heuristics. For example, AdaLoRA (Zhang et al., 2023) introduces an additional regularizer to ensure that the lower and upper projection matrices strictly adhere to the definition of singular value decomposition (SVD), with each matrix being orthogonal. However, this regularization term incurs substantial computational overhead due to the gradient calculations. In contrast, we eliminate this requirement and instead selectively filter low-rank components by controlling the intermediate diagonal matrix. We detailedly compare SoRA and related methods in Section 3.

The mechanism of SoRA also allows us to control the sparsity temporarily and investigate the relationship between the number of non-zero trainable parameters and memorization and generalization capabilities. We propose a sparsifying scheduler and find that the process of model adaptation exhibits a strong "compression capability", and even a tiny portion of parameters (lower than LoRA rank being 1) could retain considerable performance. Extensive experiments are conducted to demonstrate the effectiveness of our method. Particularly, our model could consistently outperform parameter-efficient baselines with fewer parameters and 30% shorter training time on a wide range of downstream tasks. The code of this work will be publicly available at https://github.com/TsinghuaC3I/SoRA.

## 2 A Closer Look to Adaptive Rank

**Related Work.** Before introducing our approach, we first briefly recap parameter-efficient tuning and our backbone low-rank adaptation (LoRA). Parameter-efficient tuning is a set of methods that only optimize a small portion of parameters and keep the main model untouched for adaptation. Some parameter-efficient methods would insert additional neural modules or parameters to the backbone model, such as Adapter (Houlsby et al., 2019), Prefix and Prompt Tuning (Li and Liang, 2021; Lester et al., 2021). And another line of such methods attempts to specify particular parameters to be trainable or prunable (Guo et al., 2021; Zhao et al., 2020; Zaken et al., 2021). Researchers derive a series of variants of parameter-efficient methods to improve the effectiveness or efficiency (Karimi Mahabadi et al., 2021; Hu et al., 2022; Sung et al., 2022; He et al., 2022). Recently, the applications of parameter-efficient fine-tuning are expanded to multi-modal and instruction-tuning scenarios (Gao et al., 2023; Dettmers et al., 2023). In this paper, we focus more on LoRA (Hu et al., 2021), which uses low-rank matrices to approximate the change of weights.

In LoRA, pre-trained weights (denoted as $\mathbf{W}_0 \in \mathbb{R}^{p \times q}$) are frozen, and the trainable LoRA modules are low-rank decomposition matrices $\mathbf{W}_d \in \mathbb{R}^{r \times q}$ and $\mathbf{W}_u \in \mathbb{R}^{p \times r}$ of the change of each weight matrix $\mathbf{\Delta} = \mathbf{W}_u \mathbf{W}_d \in \mathbb{R}^{p \times q}$. In this way, the output of the current layer $\mathbf{h}$ could be represented as

$$\mathbf{y} \leftarrow \mathbf{W}_0 \mathbf{x} + \mathbf{W}_u \mathbf{W}_d \mathbf{x}, \qquad (1)$$

where $r \ll \min\{p, q\}$ is a hyper-parameter of "intrinsic dimension" that controls the size of low-rank matrices and the number of trainable parameters. In this section, we primarily focus on the last term, denoting $\mathbf{z} \leftarrow \mathbf{W}_u \mathbf{W}_d \mathbf{x}$.

**Adaptive Rank on LoRA.** Despite a great step forward in tractability and efficiency, LoRA is still restricted by its inflexibility in selecting the optimal rank $r$. Unlike continuous hyperparameters such as learning rate and weight decay that can be tuned adaptively online during the training process, LoRA rank $r$ takes discrete values – the change of which will directly alter the model structures. The optimal choice of rank can vary across different backbone models and downstream tasks. A conservative choice of huge rank $r$ can waste training time and computation resources, while progressively setting $r$ tiny may degrade model performance and

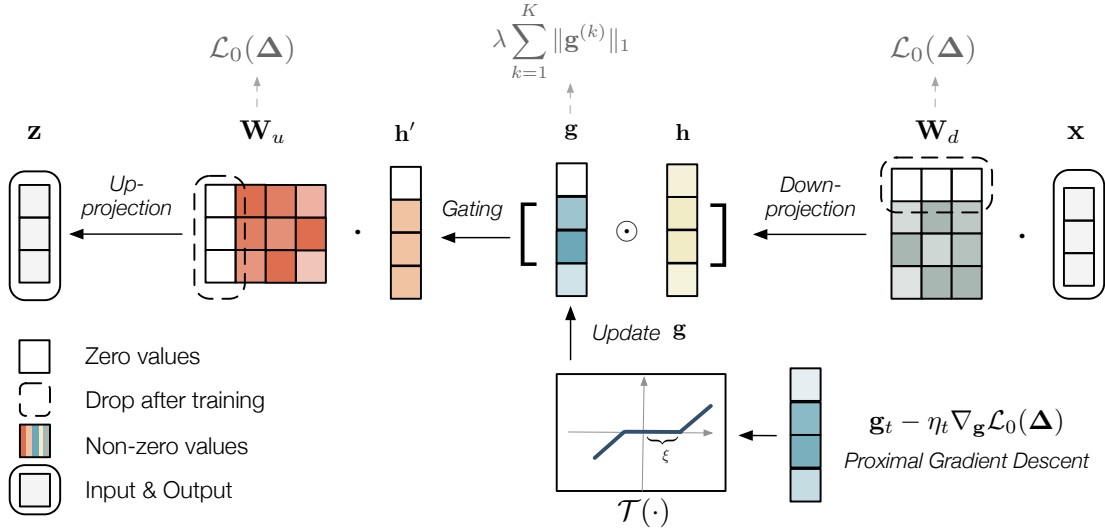

Figure 1: An illustration of sparse low-rank adaptation (SoRA). At the training stage, the gate $\mathbf{g}$ will control the sparsity of $\mathbf{W}_d$ and $\mathbf{W}_u$. At the inference stage, zero vectors in $\mathbf{W}_d$ and $\mathbf{W}_u$, indexed by the zero entries of $\mathbf{g}$, would be eliminated.

lead to from-scratch re-training. These limitations highlight the importance of upgrading LoRA with an adaptive-rank-selection plug-in.

Several remedies have been proposed in recent years to enable the flexible tuning of LoRA rank. For example, rather than setting a fixed rank, Valipour et al. (Valipour et al., 2022) introduce DyLoRA in which a pre-defined discrete distribution $p_B(\cdot)$ is cast over a range of rank choices. This approach is related to but different from nested dropout (Rippel et al., 2014), and can be regarded as optimizing a mixture model with LoRA modules of different ranks.

Nevertheless, tuning LoRA rank straightforwardly and deterministically appears to be a more attractive approach. To devise such an approach, we first gain a crucial hint from the connection between a matrix's rank and its singular value decomposition (SVD). Let us denote the tunable incremental weight matrix in LoRA by $\boldsymbol{\Delta} := \mathbf{W}_u \mathbf{W}_d$. We can then formulate its SVD as

$$\boldsymbol{\Delta}_{p \times q} = \mathbf{U}_{p \times p} \boldsymbol{\Sigma}_{p \times q} \mathbf{V}_{q \times q}^{\top}, \qquad (2)$$

in which $\mathbf{U}$ and $\mathbf{V}$ are orthogonal respectively, and $\boldsymbol{\Sigma}$ is a (rectangular) diagonal matrix with diagonal elements being the singular values of $\boldsymbol{\Delta}$: $\sigma(\boldsymbol{\Delta}) = \{\sigma_1 \geq \sigma_2 \geq \cdots \geq \sigma_{\min\{p,q\}} \geq 0\}$. For notation convenience, we reshape the diagonal of $\boldsymbol{\Sigma}$ into a column vector

$$\mathbf{g} := (\sigma_1, \sigma_2, \cdots, \sigma_{\min\{p,q\}})^{\top}. \qquad (3)$$

Then, letting $d = \min\{p, q\}$, we can reformulate the LoRA forward propagation as

$$\mathbf{z} \leftarrow \boldsymbol{\Delta}\mathbf{x} = \mathbf{U}_{.,1:d}(\mathbf{g} \odot \mathbf{V}_{.,1:d}^{\top}\mathbf{x}), \qquad (4)$$

where $\odot$ denotes element-wise dot product (Hadamard product). Note that $\text{rank}(\boldsymbol{\Delta}) = \|\mathbf{g}\|_0$ which is the $\ell_0$ norm of $\mathbf{g}$. Therefore, tuning the LoRA rank suffices to control the sparsity of the vector $\mathbf{g}$. Zhang et al. precede along this SVD-based track with their methodology named AdaLoRA (Zhang et al., 2023). In AdaLoRA, the elements in vector $\mathbf{g}$ are calibrated such that the number of nonzero entries is smaller than a pre-defined budget $b$. To be specific, they preserve only the entries with top-$b$ importance score – which is their newly proposed metric of "sensitivity" heuristically constructed from weight-gradient product. The nonnegativity of $\mathbf{g}$ entries is reasonably dropped since a negative $\mathbf{g}_i$ can be simply reduced to the positive case by flipping the sign of either $\mathbf{u}_i$ or $\mathbf{v}_i$. Besides, they transform the constrained optimization problem into its unconstrained version by replacing the orthogonality conditions $\mathbf{U}^{\top}\mathbf{U} = \mathbf{I}_p$ and $\mathbf{V}^{\top}\mathbf{V} = \mathbf{I}_q$ with a regularization term

$$R(\mathbf{U}, \mathbf{V}) = \|\mathbf{U}^{\top}\mathbf{U} - \mathbf{I}_p\|_F^2 + \|\mathbf{V}^{\top}\mathbf{V} - \mathbf{I}_q\|_F^2. \qquad (5)$$

In spite of the effectiveness demonstrated through experiments, there are still two problems in AdaLoRA that demand rethinking of the methodology and wait for further improvements. First, the

sparsity selection criterion in AdaLoRA is based on their newly proposed importance score relied on the moving average of weight-gradient product. Despite its effectiveness in empirical study, this criterion is largely heuristic, lacking theoretical motivation. Second, both the moving average operation of importance scores and the gradients of orthogonality regularization (5) add up to additional computation cost. Compared to AdaLoRA with the aforementioned limitations, our approach, SoRA, serves as an amelioration with highly simplified updating rules and is backed up by the theory of sparsity regularization and proximal gradient methods. Detailed methodology of SoRA will be elaborated in the next section.

## 3 Our Approach

The key idea of our approach, sparse low-rank adaptation (SoRA), is to dynamically adjust the intrinsic rank in the training process with a sparse gating unit trained by proximal gradient method. SoRA adopts the previously introduced framework of low-rank decomposition because of its widely validated effectiveness and parameter efficiency.

### 3.1 Sparse Low-rank Adaptation

**Module Structure.** At the start of building a SoRA module, we pre-define a maximum acceptable rank $r_{\max}$ according to practical or research concerns. Then, each SoRA module will inherit two matrices $\mathbf{W}_d \in \mathbb{R}^{r_{\max} \times q}$ and $\mathbf{W}_u \in \mathbb{R}^{p \times r_{\max}}$ from LoRA for down projection and up projection. The maximum rank $r_{\max}$ is set to be relatively large, but we will show in the subsequent paragraph how to tame it efficiently in a sparse sense. In fact, this is realized by injecting a gating unit $\mathbf{g} \in \mathbb{R}^{r_{\max}}$ between the projection matrices, which imitates the formulation of SVD. The forward propagation of the SoRA module proceeds as follows:

$$\mathbf{h} \xleftarrow{\text{down projection}} \mathbf{W}_d \mathbf{x}; \qquad (6)$$

$$\mathbf{h}' \xleftarrow{\text{gating}} \mathbf{g} \odot \mathbf{h}; \qquad (7)$$

$$\mathbf{z} \xleftarrow{\text{up projection}} \mathbf{W}_u \mathbf{h}'; \qquad (8)$$

or, more compactly,

$$\mathbf{z} \leftarrow \mathbf{W}_u \left( \mathbf{g} \odot (\mathbf{W}_d \mathbf{x}) \right). \qquad (9)$$

**Optimization.** We optimize down-projection and up-projection matrices with stochastic gradient

methods as in LoRA, while each gate $\mathbf{g}$ is updated in a different sparsity-promoting way:

$$\mathbf{g}_{t+1} \leftarrow \mathcal{T}_{\eta_t \cdot \lambda}(\mathbf{g}_t - \eta_t \nabla_{\mathbf{g}} \mathcal{L}_0(\boldsymbol{\Delta}_t)), \qquad (10)$$

in which $\mathcal{L}_0(\cdot)$ is the original loss function of the language model, $\boldsymbol{\Delta}$ denotes the complete tunable parameter (including the gates), $\eta_t > 0$ stands for the step-size at the $t$-th iteration, and $\lambda > 0$ works as the regularization strength hyperparameter that promotes sparsity. Besides, $\mathcal{T}_{\eta_t \cdot \lambda}(\cdot)$ in the above expression stands for the element-wise broadcast of the following soft-thresholding function:

$$\mathcal{T}_\xi(x) := \begin{cases} x - \xi, & x > \xi \\ 0, & -\xi < x \le \xi \\ x + \xi, & x \le -\xi \end{cases} \qquad (11)$$

with $\xi = \eta_t \cdot \lambda$ being the threshold. In practice, the true gradient $\nabla_{\mathbf{g}} \mathcal{L}_0$ in (10) is approximated by its mini-batch stochastic counterpart.

**Post-pruning.** When training is completed, we further prune the SoRA weights to drop the zeroed-out ranks and reduce the module back to the LoRA form. To be specific, for the $k$-th SoRA module, let

$$\mathcal{I}^{(k)} = \left\{ i \in [1 : r_{\max}] \mid \mathbf{g}_i^{(k)} = 0 \right\} \qquad (12)$$

be the index of zero entry in the $k$-th gating vector $\mathbf{g}^{(k)}$. We drop the $\mathcal{I}^{(k)}$-th rows of down-projection $\mathbf{W}_d^{(k)}$ to obtain $\widetilde{\mathbf{W}}_d^{(k)}$, the $\mathcal{I}^{(k)}$-th columns of up-projection $\mathbf{W}_u^{(k)}$ to obtain $\widetilde{\mathbf{W}}_u^{(k)}$, as well as the $\mathcal{I}^{(k)}$-th entry of gate $\mathbf{g}^{(k)}$ to obtain $\widetilde{\mathbf{g}}^{(k)}$. In this way, during inference time the $k$-th SoRA module will proceed as a usual LoRA module of rank $r_{\max} - |\mathcal{I}^{(k)}|$ with down-projection matrix $\widetilde{\mathbf{W}}_d^{(k)}$ and up-projection matrix $\widetilde{\mathbf{W}}_u^{(k)} \cdot \text{diag}(\widetilde{\mathbf{g}}^{(k)})$.

### 3.2 Interpretation and Comparison

**Theoretical interpretation.** The update rule (10) is in fact an application of the proximal gradient method for $\ell_1$ loss (Chambolle et al., 1998; Beck and Teboulle, 2009). This follows immediately once we reformulate (10) equivalently as

$$\mathbf{g}_{t+1} \leftarrow \arg\min_{\mathbf{g}} \ \eta_t \cdot \lambda \|\mathbf{g}\|_1$$
$$+ \frac{1}{2} \|\mathbf{g} - (\mathbf{g}_t - \eta_t \nabla \mathcal{L}_0(\mathbf{g}_t))\|_2^2. \qquad (13)$$

The above equation (13) is exactly the proximal gradient update of the $\ell_1$ regularized loss function

$$\mathcal{L}(\boldsymbol{\Delta}) := \mathcal{L}_0(\boldsymbol{\Delta}) + \lambda \sum_{k=1}^{K} \|\mathbf{g}^{(k)}\|_1, \qquad (14)$$

where $\mathbf{g}^{(k)}$ denotes the gate of the $k$-th SoRA module. This sparsity-promoting strategy dates back to LASSO estimator (Tibshirani, 1996) and compressed sensing (Candes et al., 2006), and is also adopted by many works within the realm of deep learning (Wen et al., 2016; Scardapane et al., 2017).

**Comparision with AdaLoRA.** Inspired alike by SVD decomposition, our approach SoRA differs from the preceding work AdaLoRA (Zhang et al., 2023) in the following sense. First, we do not apply the orthogonality regularization (5) used in AdaLoRA. The reason is that for rank selection purposes, sparsifying the gate $\mathbf{g}$ will be sufficient. Sticking to the original requirements of SVD can result in additional computation expenditure. Second, the moving averaged importance score in AdaLoRA works as an approximation to the change in loss when the corresponding entry is zeroed out, which is regarded as a heuristic measurement of parameter "sensitivity". However, a model's temporal sensitivity to a certain parameter cannot imply that the parameter should be retained, since there is no rigorous theory for doing so. By contrast, our rank selection based on soft-thresholding operation (10) proceeds in a much cleaner form and is soundly justified by the theory of proximal gradient iteration. As is explained earlier this section, the updating rule of SoRA module exactly follows the first principle of interpolation-complexity trade-off by minimizing a regularized loss objective (14).

Beyond the formal simplicity and theoretical clearness is SoRA's superior experimental performance achieved with fewer parameters in less wall-clock time, which will be presented in Section 4.

### 3.3 Scheduling $\xi$ to Explore Memorization and Generalization

We dub the threshold $\xi$ as a sparsity indicator. As the name implies, this parameter could directly determine the sparsity of SoRA in the training process. It can be set as a constant to heuristically control the sparsity according to the budget of parameters and expected performance. When dynamically changing $\xi$ in the adaptation process, SoRA serves as an effective tool to assess the memorization and generalization under a model $\mathcal{M}$ and a dataset $\mathcal{D}$. In other words, we can visually observe how many additional parameters are required to achieve a particular point of performance given the model $\mathcal{M}$ and data $\mathcal{D}$. We elaborate the fundamental idea as follows. The process starts by assigning a rel-

atively small value to $\xi$. Consequently, the SoRA model is initially "dense" and is trained until convergence. Once this stage is achieved, we introduce a scheduler to incrementally increase the value of $\xi$, thereby enhancing the model's sparsity. During this transition from a dense to a sparse model, it becomes possible to evaluate the model's memorization and generalization abilities by examining performance on the training and testing data respectively. The procedure is reported in Algorithm 1.

The process can be regarded as exploring the "compression loss" in the scenario of model adaptation. Here, "compression loss" refers to the reduction in model performance due to the increased sparsity, providing a measure of how well the model can retain its predictive power under constraints. Investigating this "compression loss" is meaningful to understanding the behavior of model adaptation and can facilitate developing efficient, compact models that maintain high-performance levels.

---

**Algorithm 1:** Scheduling Algorithm of $\xi$

**Input** : $\mathcal{M}, \xi_0, \xi_{\max}, \delta_\xi, \mathcal{D}$
**Output** : $\mathcal{M}' = \{\mathcal{M}_0, \mathcal{M}_1, ...\}$
$\xi \leftarrow \xi_0$;
$\mathcal{M}' \leftarrow \emptyset$;
$\mathcal{M} = TrainUntilConvergence(\mathcal{M}, \mathcal{D}, \xi)$;
$\mathcal{M}'.add(\mathcal{M})$;
$\xi \leftarrow \xi + \xi_\lambda$;
**while** $\xi \leq \xi_{max}$ **do**
  **for** $epoch \leftarrow 1$ **to** $5$ **do**
   | $\mathcal{M} = Update(\mathcal{M}, \mathcal{D}, \xi)$;
  **end**
  $\mathcal{M}'.add(\mathcal{M})$;
  $\xi \leftarrow \xi + \delta_\xi$;
**end**

---

## 4 Experiments

Extensive experiments are carried out to assess the effectiveness of our approach comprehensively. Generally speaking, we explore two aspects in this section: (1) the performance and corresponding analysis as a normal parameter-efficient method; and (2) the investigation of memorization and generalization in virtue of the sparsity nature of SoRA.

### 4.1 Experimental Settings.

**Baselines.** Our baselines comprise full-parameter fine-tuning and other well-recognized parameter-efficient methods, including Adapter (Houlsby

| Method | #Params | CoLA | SST-2 | MRPC | QQP | STS-B | MNLI | QNLI | RTE | Avg. |
|---|---|---|---|---|---|---|---|---|---|---|
| Fine-Tune | 184M | 69.21 | 95.64 | 89.22 | 92.05/89.31 | 91.59 | 89.98/89.95 | 93.78 | 82.49 | 87.82 |
| Adapter | 1.41M | 69.00 | 95.16 | 89.90 | 91.45/88.88 | 92.21 | 90.11/90.11 | 93.79 | 82.44 | 87.85 |
| Bitfit | 0.1M | 68.70 | 94.38 | 87.16 | 87.86/84.20 | 89.71 | 87.45/87.45 | 91.90 | 76.12 | 85.18 |
| LoRA (r=8) | 1.33M | 69.73 | 95.57 | 89.71 | 91.95/89.26 | 91.86 | **90.47/90.46** | 93.76 | 85.32 | 88.38 |
| AdaLoRA | 1.27M | 70.86 | **95.95** | 90.22 | 92.13/88.41 | 91.39 | 90.27/90.30 | **94.28** | 87.36 | 88.83 |
| SoRA | 0.91M | **71.48** | 95.64 | **91.98** | **92.39/89.87** | **92.22** | 90.35/90.38 | **94.28** | **87.77** | **89.36** |

Table 1: Test results of SoRA and other baselines on the GLUE benchmark. We denote the best result in **bold** and underline the second best result. The standard deviations of results from different methods are similar and we show them in Table 8 in Appendix A.4.

et al., 2019), BitFit (Zaken et al., 2021), LoRA (Hu et al., 2021) and AdaLoRA (Zhang et al., 2023). We omit the variants of the Adapter since we find that the performance between them is very close. We also do not include Prompt Tuning since we find that it takes considerably longer time for convergence and cannot yield non-trivial performance on our backbone models.

**Datasets** For evaluation, we adaopt the GLUE benchmark (Wang et al.), including CoLA (Warstadt et al., 2019), SST-2 (Socher et al., 2013), MRPC (Dolan and Brockett, 2005), QQP (Wang et al.), STS-B (Wang et al.), MNLI (Williams et al., 2017), QNLI (Rajpurkar et al., 2016) and RTE (Dagan et al., 2005; Haim et al., 2006; Giampiccolo et al., 2007; Bentivogli et al., 2009). We mainly use DeBERTaV3-base (He et al., 2021) as the backbone model. Additionally, we also use RoBERTa-large (Liu et al., 2019) for analysis. Other experimental details are described in Appendix A.

## 4.2 Results

We first conduct an evaluation on GLUE benchmark, a widely recognized benchmark for natural language understanding. The experimental performance of SoRA, as well as other baseline methodologies, is recorded in Table 1. We reproduce these methods in our infrastructure and present the average results drawn from 5 random seeds. Our findings indicate that both AdaLoRA and SoRA consistently outperform the initial LoRA baseline. This underlines the validity of adaptive rank as a potent solution for enhanced model adaptation. Most notably, SoRA outshines all other baselines, particularly LoRA and AdaLoRA, despite utilizing fewer parameters. This lends credence to the argument that our proximal gradient method may constitute a more efficacious and essential approach to achieving adaptive rank. For instance, on the MRPC,

SoRA achieved an accuracy of 91.98%, surpassing AdaLoRA by 1.76%. On average, SoRA surpassed LoRA and AdaLoRA on the GLUE benchmark by 0.98% and 0.52%, respectively, using 31.5% and 28.3% fewer parameters. To take a closer look at the effectiveness of adaptive rank, we conduct an experiment to compare LoRA and SoRA with different ranks in Table 2. The results affirm that SoRA's superiority is consistent across different budgets of parameters, that is, SoRA could outperform the LoRA baseline in all settings while utilizing over 30% fewer parameters.

## 4.3 Sparsifying Scheduler

We apply the sparsifying scheduler introduced in Section 3.3 by enlarging the sparse indicator $\xi$ (starting from 1e-4) of SoRA progressively in the adaptation process. As illustrated in Figure 2, we plot the memorization and generalization curve of RoBERTa-large (Liu et al., 2019) on MRPC, RTE, STS-B, CoLA, QNLI, and SST-2, where the memorization is gauged by the performance on the training set and the generalization is measured by the performance on the validation set. Intriguingly, we observe a robust "compression performance" across almost all the datasets. Among these, SST-2 emerges as the most "compressible" task, where the model sustains over 99% performance even when restricted to 47,104 non-zero parameters. Remarkably, a mere 4,096 parameters can still conserve above 90% memorization and generalization capabilities. As the sparsifying process proceeds, the model encounters an "inflection point" on different data, after which the performance significantly plummets. This consistent phenomenon suggests that there exist some critical parameters that underpin the performance and are worth further investigation. Insight gleaned from the graph also indicates varying degrees of adaptation difficulty for the model across different datasets. For example, certain datasets, like CoLA, prompt an earlier

| Method | #Params | CoLA | SST-2 | MRPC | QQP | STS-B | MNLI | QNLI | RTE | Avg. |
|---|---|---|---|---|---|---|---|---|---|---|
| $\text{LoRA}_{r=1}$ | 0.17M | 68.60 | 94.95 | 88.24 | 91.20/88.37 | 91.41 | 90.09/90.28 | 93.35 | 81.29 | 87.23 |
| $\text{SoRA}_{r=1}$ | $0.12\text{M}_{-29.41\%}$ | $70.24_{+1.64}$ | $95.14_{+0.19}$ | $89.22_{+0.98}$ | $91.52/88.73_{+0.34}$ | $91.41_{+0.00}$ | $90.08/90.41_{+0.06}$ | $93.43_{+0.08}$ | $83.02_{+1.73}$ | $87.85_{+0.62}$ |
| $\text{LoRA}_{r=2}$ | 0.33M | 68.93 | 95.04 | 88.43 | 91.59/88.87 | 91.53 | 90.35/90.30 | 93.63 | 84.17 | 87.79 |
| $\text{SoRA}_{r=2}$ | $0.25\text{M}_{-24.24\%}$ | $70.22_{+1.29}$ | $95.64_{+0.60}$ | $89.71_{+1.28}$ | $91.88/89.12_{+0.27}$ | $91.63_{+0.10}$ | $90.37/90.51_{+0.12}$ | $93.78_{+0.15}$ | $85.18_{+1.01}$ | $88.39_{+0.60}$ |
| $\text{LoRA}_{r=4}$ | 0.66M | 69.27 | 95.55 | 89.22 | 91.40/88.41 | 91.69 | **90.36/90.49** | 93.83 | 82.01 | 87.74 |
| $\text{SoRA}_{r=4}$ | $0.47\text{M}_{-28.79\%}$ | $\textbf{71.05}_{+1.78}$ | $95.57_{+0.02}$ | $90.20_{+0.98}$ | $92.06/89.44_{+0.85}$ | $91.76_{+0.07}$ | $90.38/90.43_{-0.02}$ | $93.92_{+0.09}$ | $86.04_{+4.03}$ | $88.71_{+0.97}$ |
| $\text{LoRA}_{r=8}$ | 1.33M | 69.73 | 95.57 | 89.71 | 91.95/89.26 | 91.86 | **90.47/90.46** | 93.76 | 85.32 | 88.38 |
| $\text{SoRA}_{r=8}$ | $0.91\text{M}_{-31.58\%}$ | $\textbf{71.48}_{+1.75}$ | $95.64_{+0.07}$ | $91.98_{+2.27}$ | $92.39/89.87_{+0.53}$ | $92.22_{+0.36}$ | $90.35/90.38_{-0.10}$ | $94.28_{+0.52}$ | $87.77_{+2.45}$ | $89.36_{+0.98}$ |
| $\text{LoRA}_{r=16}$ | 2.65M | 69.87 | 95.53 | 89.91 | 92.22/89.63 | 91.79 | **90.55/90.31** | 93.46 | 87.05 | 88.62 |
| $\text{SoRA}_{r=16}$ | $1.78\text{M}_{-32.83\%}$ | $\textbf{71.93}_{+2.06}$ | $95.61_{+0.08}$ | $92.00_{+2.09}$ | $92.37/89.84_{+0.18}$ | $92.05_{+0.26}$ | $90.34/90.47_{-0.03}$ | $94.11_{+0.65}$ | $87.41_{+0.36}$ | $89.33_{+0.71}$ |

Table 2: Test results and number of parameters of SoRA initialized with different $r_{\max}$ on the GLUE benchmark, compared with LoRA of the same rank. The standard deviations of results from different methods are similar and we show them in Table 9 in Appendix A.4.

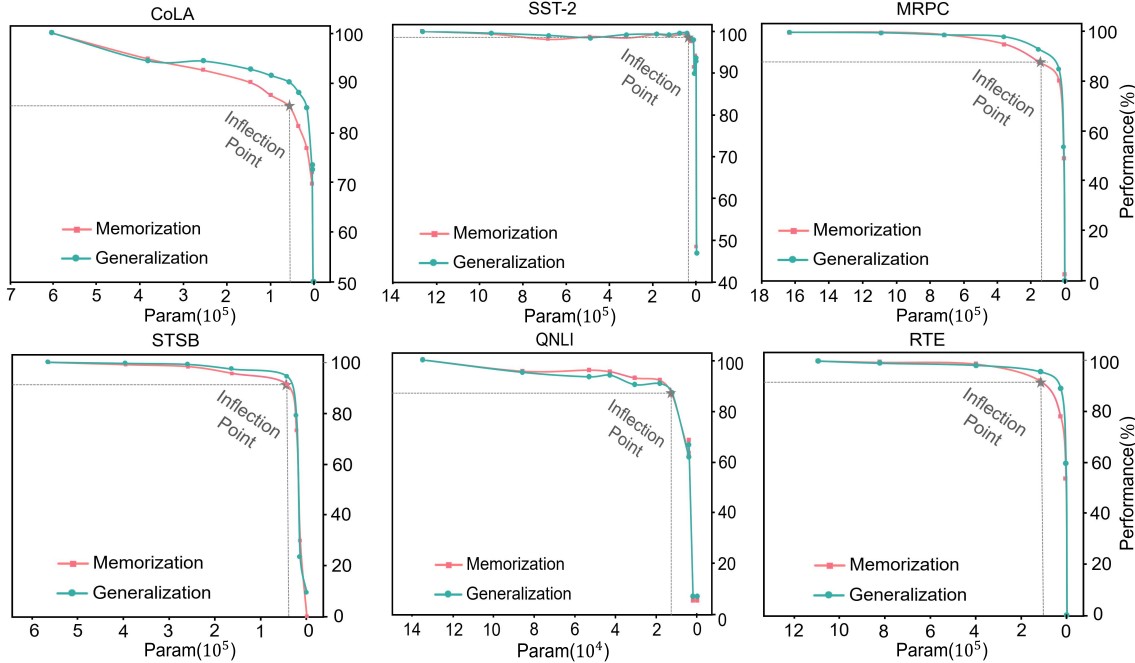

Figure 2: The memorization and generalization curve on six datasets. The "Param" axis indicates the number of non-zero parameters. The sparsity indicator $\xi$ increases every 5 epochs.

and more pronounced decline in performance compared to others. Another finding is that the trend of memorization and generalization is consistent in the sparsifying procedure, which is aligned with intuition. Our observations also indicate a tendency for the parameters of intermediate and deep layers to maintain their density, while those of the shallow layers show a higher propensity towards sparsity.

### 4.4 Rank Analysis

An intuitive statement is that a single model suffers from varying extents of difficulty when being adapted to different downstream datasets. Concurrently, it is evident that not all parameters within the model carry equal importance—some are more critical to performance than others. In this section, we visualize the final ranks after the training process

converges with SoRA on four datasets in Figure 3. Quite obviously, the trained parameter matrices on QQP are exceedingly dense and others do not exhibit such density, which echos the existence of different levels of difficulties. This phenomenon also suggests that leveraging the performance and the parameter budget does not have an invariable constant law, but needs specific considerations in different situations.

### 4.5 Applying SoRA to Different Weights

In our experiments in Table 1, we utilize LoRA, AdaLoRA, and SoRA on all weight matrices to enhance performance. It should be noted that the performance may fluctuate when parameter-efficient fine-tuning is applied to various positions within the model, as evidenced by previous

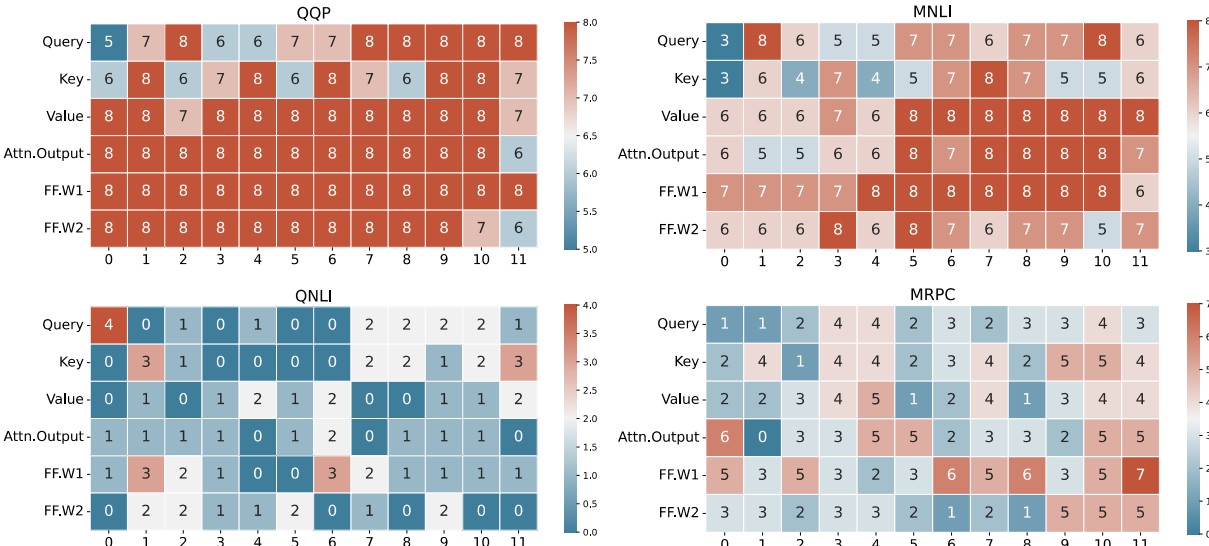

Figure 3: The final ranks after training with SoRA on four datasets (l.e., QQP, MNLI, QNLI, and MRPC). The X-axis is the index of DeBERTaV3-base layers, and the Y-axis indicates different layers SoRA applies to.

research (Zaken et al., 2021; Hu et al., 2022; Zhang et al., 2023). We carry out such ablation experiments with SoRA on three datasets to investigate the impact. Although SoRA is not a budget-oriented method, we adjust $\lambda$ to approximately equate the retained non-zero parameters. As reported in Table 3, in most cases, the application of SoRA to all weight matrices resulted in a considerable improvement in performance compared to the application of merely one or several types of weights, which suggest that uniformly applying SoRA to all weight matrices can serve as a beneficial strategy. And merely applying SoRA to $\mathbf{W}_{Q,K}$ will experience considerable performance drop, which is aligned with LoRA.

|  | #Params | CoLA | MRPC | STS-B |
|---|---|---|---|---|
| $\mathbf{W}_{Q,K}$ | 0.80M | $65.56_{\pm2.24}$ | $86.43_{\pm0.83}$ | $72.30_{\pm2.61}$ |
| $\mathbf{W}_{Q,K,V}$ | 0.69M | $69.07_{\pm2.17}$ | $88.24_{\pm1.84}$ | $90.82_{\pm0.70}$ |
| $\mathbf{W}_{Q,K,V,A.O}$ | 0.87M | $71.99_{\pm1.30}$ | $90.20_{\pm1.83}$ | $91.71_{\pm0.34}$ |
| All | 0.77M | $71.48_{\pm1.17}$ | $91.98_{\pm1.16}$ | $92.22_{\pm0.24}$ |

Table 3: Test results that applying SoRA to different weights. Q, K, V, and A.O represent query, key, value and attention output layers respectively. **#Params** means the number of parameters that would remain after training.

### 4.6 Efficiency Analysis

We elaborate that SoRA is a theoretically clear and computation-efficient method in Section 3.2. To evaluate this, we measure the efficiency of SoRA and AdaLora in this section. We compute the clock time of average epoch of AdaLoRA and SoRA on six datasets with identical compute infrastructure

and batch size. As shown in Table 4, SoRA takes about 30% less training time than AdaLoRA. In certain instances, such as the CoLA, QNLI, and RTE datasets, SoRA exhibits a significant edge in efficiency over its counterpart. Conversely, while SoRA consistently outpaces AdaLoRA on other datasets, the margin is not as wide. This discrepancy could be attributable to the different rank distributions of AdaLoRA and SoRA under varying tasks. Such distributions exert influence on the calculation of regularization in AdaLoRA.

| Datasets | AdaLoRA (s) | SoRA (s) |
|---|---|---|
| CoLA | 160.2 | $57.2_{-64.29\%}$ |
| SST-2 | 491.0 | $433.0_{-11.81\%}$ |
| MRPC | 27.3 | $24.8_{-9.16\%}$ |
| STS-B | 48.2 | $38.4_{-20.33\%}$ |
| QNLI | 1001.0 | $676.3_{-32.44\%}$ |
| RTE | 79.8 | $45.1_{-43.48\%}$ |
| Avg. | 301.3 | $212.5_{-29.47\%}$ |

Table 4: The average training time per epoch on six datasets. For each task, the experiments with AdaLoRA and SoRA have the same batch size 32.

### 5 Conclusion

Our work presents Sparse Low-Rank Adaptation (SoRA), an innovative method for parameter-efficient fine-tuning large pre-trained language models. Upon the hypothesis that the adaptation process could be intrinsically sparse, we offer a dynamic alternative rank by introducing an optimizable gate with a proximal gradient method to

regulate sparsity, thereby expanding the optimization space while enhancing parameter efficiency. The method is simple and theoretically supported with promising performance across various tasks. Utilizing SoRA as a tool, we propose a sparsifying scheduler to analyze the correlation between parameters and memorization and generalization.

## Limitations

Despite the encouraging results demonstrated by SoRA, there are certain limitations in our current study that are worth acknowledging. This paper only evaluates the effectiveness of SoRA on traditional natural language processing tasks. However, recent studies demonstrate that parameter-efficient methods could be applied to cross-modal or instruction-tuning scenarios. In those cases, how the sparsity of SoRA is displayed is still unknown and worth investigating. Our sparsifying scheduler could provide insights on the adaptation process of language models, but it is still challenging to rigorously explain the procedure and more efficiently to assess the difficulty of an adaptation process.

## Acknowledgements

This work is supported by the National Key R&D Program of China (No. 2022ZD0119101), National Natural Science Foundation of China (No. 62236004), the Young Elite Scientists Sponsorship Program by CAST, and Institute Guo Qiang at Tsinghua University.

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

## A Experimental Details

### A.1 Datasets

The GLUE benchmark, consisting of CoLA (Warstadt et al., 2019), SST-2 (Socher et al., 2013), MRPC (Dolan and Brockett, 2005), QQP (Wang et al.), STS-B (Wang et al.), MNLI (Williams et al., 2017), QNLI (Rajpurkar et al., 2016) and RTE (Dagan et al., 2005; Haim et al., 2006; Giampiccolo et al., 2007; Bentivogli et al., 2009), is used for natural language understanding. The details and the evaluation metric are reported in Table 5. We source each dataset from Huggingface Datasets (Lhoest et al., 2021) and utilize the full dataset for our experiments. For almost all experiments, we run 5 times using different random seeds and report the average results in order to ensure statistical significance.

| Dataset | #Train | #Valid | #Test | Metric |
|---------|--------|--------|-------|--------|
| CoLA | 8.5k | 1,043 | 1,063 | Mcc |
| SST-2 | 67k | 872 | 1.8k | Acc |
| MRPC | 3.7k | 408 | 1.7k | Acc |
| QQP | 364k | 40.4k | 391k | Acc/F1 |
| STS-B | 5.7k | 1.5k | 1.4k | Corr |
| MNLI | 393k | 9.8k/9.8k | 9.8k/9.8k | Acc(m/mm) |
| QNLI | 105k | 5.5k | 5.5k | Acc |
| RTE | 2.5k | 277 | 3k | Acc |

Table 5: The size and evaluation metric of the datasets in GLUE benchmark. "Mcc", "Acc", "F1" and "Corr" represent matthews correlation coefficient, accuracy, the F1 score and pearson correlation coefficient respectively. And "Acc(m/mm)" represents the results corresponding to matched and mismatched datasets of MNLI while the metric is accuracy.

### A.2 Implementation Details

Regarding hyper-parameters, we set the learning rate to 8e-4. Based on the size and training convergence speed of the datasets, we set the number of epochs for CoLA, MRPC, and STS-B to 20, and the number of epochs for the remaining tasks to 10. As for RTE, we reference the settings of Friedman et al. 2021, which entail a learning rate of 1.2e-3 and an epoch count of 50. We set $\lambda$ to 0.1 in all our experiments, and select $\xi$ with a grid search in {1e-5, 5e-5, 1e-4}. When dealing with MRPC, RTE, and STS-B datasets, a common trick in certain studies is that using the best model checkpoint on the MNLI dataset could boost the performance. In our experiments, we do not use this strategy and instead opt for standard initializations across all models.

The Huggingface Transformers (Wolf et al., 2020) and PyTorch (Paszke et al., 2019) are utilized for all the experiments. We use NVIDIA GeForce RTX 3090 (maximum GPU memory=24268MB) and the application of SoRA with a batch size of 8 occupies 6110MB GPU memory on average.

### A.3 Optimization of Hyperparameters

In this section, we delve into the optimization of hyperparameters. The results of different $r_{\max}$ are proved in Table 2 and we supplement the results of two other important hyperparameters, $\xi$ and $\eta$ in the Table 6 and Table 7. The performance of SoRA is highly stable with respect to different choices of $\xi$ and $\eta$. And for each fixed $\xi$ and $\eta$, the variance of performance is rather low. In general, we suggest setting $\xi$ to 1e-4 level and $\eta$ around 1e-1∼1e-3.

| $\xi$ | COLA | STS-B |
|-------|------|-------|
| 1e-3 | $63.25_{\pm 0.71}$ | $90.85_{\pm 0.53}$ |
| 8e-4 | $64.98_{\pm 1.25}$ | $91.12_{\pm 0.38}$ |
| 5e-4 | $66.94_{\pm 0.98}$ | $91.61_{\pm 0.24}$ |
| 3e-4 | $68.61_{\pm 1.23}$ | $92.22_{\pm 0.24}$ |
| 1e-4 | $70.18_{\pm 1.05}$ | $92.01_{\pm 0.14}$ |
| 8e-5 | $68.78_{\pm 1.43}$ | $92.00_{\pm 0.15}$ |
| 5e-5 | $71.48_{\pm 1.17}$ | $92.02_{\pm 0.18}$ |
| 3e-5 | $69.68_{\pm 1.94}$ | $92.18_{\pm 0.15}$ |
| 1e-5 | $69.65_{\pm 1.93}$ | $92.08_{\pm 0.15}$ |

Table 6: Test results of the optimization experiments on different $\xi$.

| $\eta_t$ | COLA | STS-B |
|----------|------|-------|
| 10 | $69.06_{\pm 2.18}$ | $91.75_{\pm 0.41}$ |
| 1 | $70.54_{\pm 1.15}$ | $92.02_{\pm 0.20}$ |
| 0.1 | $71.48_{\pm 1.17}$ | $92.22_{\pm 0.24}$ |
| 0.01 | $68.78_{\pm 2.29}$ | $92.06_{\pm 0.17}$ |
| 0.001 | $69.86_{\pm 1.54}$ | $91.83_{\pm 0.22}$ |
| 0.0001 | $69.70_{\pm 1.70}$ | $92.13_{\pm 0.40}$ |

Table 7: Test results of the optimization experiments on different $\eta_t$.

### A.4 Results with Standard Deviations

The test results in Table 1 are shown in Table 8, and results in Table 2 are shown in Table 9.

| Method | #Params | CoLA | SST-2 | MRPC | QQP | STS-B | MNLI | QNLI | RTE | Avg. |
|---|---|---|---|---|---|---|---|---|---|---|
| Fine-Tune | 184M | 69.21 (2.24) | 95.64 (0.52) | 89.22 (0.69) | 92.05/89.31 (0.09)/(0.07) | 91.59 (0.47) | 89.98/89.95 (0.06)/(0.33) | 93.78 (0.02) | 82.49 (1.48) | 87.82 |
| Adapter | 1.41M | 69.00 (0.91) | 95.16 (0.46) | 89.90 (2.10) | 91.45/88.88 (0.18)/(0.40) | 92.21 (0.33) | 90.11/90.11 (0.57)/(0.57) | 93.79 (0.07) | 82.44 (1.74) | 87.85 |
| Bitfit | 0.1M | 68.70 (1.85) | 94.38 (0.28) | 87.16 (0.58) | 87.86/84.20 (0.52)/(0.74) | 89.71 (0.58) | 87.45/87.45 (0.76)/(0.76) | 91.90 (0.14) | 76.12 (1.54) | 85.18 |
| LoRA (r=8) | 1.33M | 69.73 (1.42) | 95.57 (0.21) | 89.71 (1.32) | 91.95/89.26 (0.12)/(0.18) | 91.86 (0.29) | **90.47/90.46** **(0.23)/(0.12)** | 93.76 (0.36) | 85.32 (0.86) | 88.38 |
| AdaLoRA | 1.27M | 70.86 (1.43) | **95.95** **(0.37)** | 90.22 (0.40) | 92.13/88.41 (0.06)/(0.05) | 91.39 (0.25) | 90.27/90.30 (0.11)/(0.18) | **94.28** **(0.11)** | 87.36 (0.30) | 88.83 |
| SoRA | 0.91M | **71.48** **(1.17)** | 95.64 (0.23) | **91.98** **(1.16)** | **92.39/89.87** **(0.17)/(0.27)** | **92.22** **(0.24)** | 90.35/90.38 (0.09)/(0.12) | **94.28** **(0.06)** | **87.77** **(1.56)** | **89.36** |

Table 8: Test results of SoRA and other baselines on the GLUE benchmark. We denote the best result in **bold** and underline the second best result. The standard deviation is provided in parentheses.

| Method | #Params | CoLA | SST-2 | MRPC | QQP | STS-B | MNLI | QNLI | RTE | Avg. |
|---|---|---|---|---|---|---|---|---|---|---|
| $LoRA_{r=1}$ | 0.17M | 68.60 (1.58) | 94.95 (0.20) | 88.24 (1.42) | 91.20/88.37 (0.20/0.33) | 91.41 (0.29) | 90.09/90.28 (0.16/0.14) | 93.35 (0.34) | 81.29 (2.83) | 87.23 |
| $SoRA_{r=1}$ | 0.12M | **70.24** **(1.10)** | **95.14** **(0.25)** | **89.22** **(2.12)** | **91.52/88.73** **(0.14/0.22)** | 91.41 (0.29) | **90.08/90.41** **(0.18/0.12)** | **93.43** **(0.33)** | **83.02** **(2.47)** | **87.85** |
| $LoRA_{r=2}$ | 0.33M | 68.93 (1.50) | 95.04 (0.37) | 88.43 (1.06) | 91.59/88.87 (0.06/0.12) | 91.53 (0.32) | 90.35/90.30 (0.13/0.17) | 93.63 (0.37) | 84.17 (1.20) | 87.79 |
| $SoRA_{r=2}$ | 0.25M | **70.22** **(1.03)** | **95.64** **(0.09)** | **89.71** **(0.81)** | **91.88/89.12** **(0.13/0.17)** | **91.63** **(0.30)** | **90.37/90.51** **(0.06/0.08)** | **93.78** **(0.23)** | **85.18** **(0.73)** | **88.39** |
| $LoRA_{r=4}$ | 0.66M | 69.27 (2.22) | 95.55 (0.56) | 89.22 (1.64) | 91.40/88.41 (0.17/0.27) | 91.69 (0.24) | **90.36/90.49** **(0.12/0.32)** | 93.83 (0.35) | 82.01 (1.76) | 87.74 |
| $SoRA_{r=4}$ | 0.47M | **71.05** **(0.74)** | **95.57** **(0.05)** | **90.20** **(1.60)** | **92.06/89.44** **(0.17/0.14)** | **91.76** **(0.17)** | 90.38/90.43 (0.04/0.25) | **93.92** **(0.09)** | **86.04** **(1.96)** | **88.71** |
| $LoRA_{r=8}$ | 1.33M | 69.73 (1.42) | 95.57 (0.21) | 89.71 (1.32) | 91.95/89.26 (0.12/0.18) | 91.86 (0.29) | **90.47/90.46** **(0.23/0.12)** | 93.76 (0.36) | 85.32 (0.86) | 88.38 |
| $SoRA_{r=8}$ | 0.91M | **71.48** **(1.17)** | **95.64** **(0.23)** | **91.98** **(1.16)** | **92.39/89.87** **(0.17/0.27)** | **92.22** **(0.24)** | 90.35/90.38 (0.09/0.12) | **94.28** **(0.06)** | **87.77** **(1.56)** | **89.36** |
| $LoRA_{r=16}$ | 2.65M | 69.87 (0.86) | 95.53 (0.15) | 89.91 (1.69) | 92.22/89.63 (0.05/0.04) | 91.79 (0.16) | **90.55/90.31** **(0.10/0.03)** | 93.46 (0.12) | 87.05 (3.11) | 88.62 |
| $SoRA_{r=16}$ | 1.78M | **71.93** **(0.97)** | **95.61** **(0.11)** | **92.00** **(0.23)** | **92.37/89.84** **(0.15/0.19)** | **92.05** **(0.16)** | 90.34/90.47 (0.13/0.04) | **94.11** **(0.07)** | **87.41** **(1.08)** | **89.33** |

Table 9: Test results and number of parameters of SoRA initialized with different $r_{\max}$ on the GLUE benchmark, compared with LoRA of the same rank. The standard deviation is provided in parentheses.