# OpenReview forum: "Sparse Low-rank Adaptation of Pre-trained Language Models"
_EMNLP/2023/Conference — EMNLP 2023 Main_

### Official Review · Reviewer_mFGF · 2023-07-21

**Typos Grammar Style And Presentation Improvements:** Your algorithm should also use $\topr…
**Soundness:** 4

**Excitement:**

4: Strong: This paper deepens the understanding of some phenomenon or lowers the barriers to an existing research direction.

**Paper Topic And Main Contributions:**

The paper proposes a method called sparse low-rank adaptation (SoRA) for parameter-efficient fine-tuning of large pre-trained language models. SoRA introduces a sparse gating unit that dynamically adjusts the intrinsic rank of the low-rank decomposition matrices during the training process, using a proximal gradient method with sparsity regularization. It improves the representation power and parameter efficiency of the existing low-rank adaptation (LoRA) method, and outperforms other baselines on various natural language understanding tasks. It also enables the exploration of the relationship between the number of non-zero parameters and the memorization and generalization capabilities of the model, using a sparsifying scheduler.

**Questions For The Authors:**

1. How is $r_{max}$ defined? If it's set during initialization, shouldn't hyperparameter optimization be conducted?
2. Does $\eta_t$ being larger than 0 or $\eta_t > 0$ represent the step size at the t-th iteration?

**Reasons To Accept:**

(+) The author presents a fresh approach known as SORA, designed to enhance the representational strength of LoRA by initializing it at a higher rank.
(+) The paper provides a lucid theoretical explanation, detailing the underlying principles of SORA.

**Reasons To Reject:**

(-) The study falls short in terms of ablation analysis and optimization of hyperparameters.
(-) The problem definition could benefit from a clearer explanation; the extensive use of symbols can sometimes lead to confusion.
(-) The experiment should compute the mean and variance.

**Reproducibility:**

4: Could mostly reproduce the results, but there may be some variation because of sample variance or minor variations in their interpretation of the protocol or method.

**Reviewer Confidence:**

5: Positive that my evaluation is correct. I read the paper very carefully and I am very familiar with related work.

---

> ### Author Rebuttal · Authors · 2023-08-29
>
> Thank you for providing your valuable and constructive feedback.
> > The study falls short in terms of ablation analysis and optimization of hyperparameters.
> - Thanks for your comment. Our method, SoRA, is kind of an integrated strategy for parameter-efficient fine-tuning rather than a modularized method, and it is counter-intuitive to eliminate any parts of it. For example, if we remove the proximal gradient part, the method itself will not hold.
> - We agree that the effect of hyperparameters should be taken into consideration. In Table 2, we prove the results of different r_max. Now we have supplemented the results of two other important hyperparameters, $\xi$ and $\eta$ in the following tables.
> - From the following tables, we discover that the performance of SoRA is highly stable with respect to different choices of $\xi$ and $\eta$. And for each fixed $\xi$ and $\eta$, the variance of performance is rather low. In general, we suggest setting $\xi$ to 1e-4 level and $\eta$ around 1e-1 ~ 1e-3. The analysis would be updated in the revision.
>
> | $\xi$ |     CoLA     |    STS-B     |
> | :---: | :----------: | :----------: |
> | 1e-3  | 63.25 (0.71) | 90.85 (0.53) |
> | 8e-4  | 64.98 (1.25) | 91.12 (0.38) |
> | 5e-4  | 66.94 (0.98) | 91.61 (0.24) |
> | 3e-4  | 68.61 (1.23) | 92.22 (0.24) |
> | 1e-4  | 70.18 (1.05) | 92.01 (0.14) |
> | 8e-5  | 68.78 (1.43) | 92.00 (0.15) |
> | 5e-5  | 71.48 (1.17) | 92.02 (0.18) |
> | 3e-5  | 69.68 (1.94) | 92.18 (0.15) |
> | 1e-5  | 69.65 (1.93) | 92.08 (0.15) |
>
>
> | $\eta_t$ |     CoLA     |    STS-B     |
> | :------: | :----------: | :----------: |
> |    10    | 69.06 (2.18) | 91.75 (0.41) |
> |    1     | 70.54 (1.15) | 92.02 (0.20) |
> |   0.1    | 71.48 (1.17) | 92.22 (0.24) |
> |   0.01   | 68.78 (2.29) | 92.06 (0.17) |
> |  0.001   | 69.86 (1.54) | 91.83 (0.22) |
> |  0.0001  | 69.70 (1.70) | 92.13 (0.40) |
>
> > The problem definition could benefit from a clearer explanation; the extensive use of symbols can sometimes lead to confusion.
> - We apologize for the potential confusion. We find it hard to narrow down our passage since most of the background introductions and mathematical definitions are vital to readers who are new in this field. However, it is our pleasure to explain our idea more briefly here to you. To be concise, our goal is to develop a LoRA amelioration with optimizable ranks to achieve an automatic perfection of both high performance and low cost. For the symbolic issues, readers may focus on Eq 9 and Eq 10, which are all of our ideas. We will definitely try to reduce the symbol use and make the method clearer, thanks!
>
> > The experiment should compute the mean and variance.
> - Thank you for the reminder.  All results are the mean results already. We have supplemented the standard deviation and displayed it in the table below.We will also include it in the next version of the paper. The standard deviation corresponding to SoRA is not significantly different from other methods, indicating that the improvement in performance on the GLUE tasks is indeed attributed to the SoRA method.
>
> - **Table 1**
>
> |   Method   | #Params |        CoLA         |        SST-2        |        MRPC         |                QQP                |        STS-B        |               MNLI                |        QNLI         |         RTE         |     Avg.     |
> | :--------: | :-----: | :-----------------: | :-----------------: | :-----------------: | :-------------------------------: | :-----------------: | :-------------------------------: | :-----------------: | :-----------------: | :----------: |
> | Fine-Tune  |  184M   |    69.21 (2.24)     |    95.64 (0.52)     |    89.22 (0.69)     | 92.05 (0.09)/ 89.31 (0.07)|    91.59 (0.47)     |    89.98 (0.06)/ 89.95 (0.33)     |    93.78 (0.02)     |    82.49 (1.48)     |    87.82     |
> |  Adapter   |  1.41M  |    69.00 (0.91)     |    95.16 (0.46)     |    89.90 (2.10)     |    91.45 (0.18)/ 88.88 (0.40)     | 92.21 (0.33)|    90.11 (0.57)/ 90.11 (0.57)     | 93.79 (0.07)|    82.44 (1.74)     |    87.85     |
> |   Bitfit   |  0.1M   |    68.70 (1.85)     |    94.38 (0.28)     |    87.16 (0.58)     |    87.86 (0.52)/ 84.20 (0.74)     |    89.71 (0.58)     |    87.45 (0.76)/ 87.45 (0.76)     |    91.90 (0.14)     |    76.12 (1.54)     |    85.18     |
> | LoRA (r=8) |  1.33M  |    69.73 (1.42)     |    95.57 (0.21)     |    89.71 (1.32)     |    91.95 (0.12)/ 89.26 (0.18)     |    91.86 (0.29)     |  **90.47 (0.23)/ 90.46 (0.12 )**  |    93.76 (0.36)     |    85.32 (0.86)     |    88.38     |
> |  AdaLoRA   |  1.27M  | 70.86 (1.43)|  **95.95 (0.37)**   | 90.22 (0.40) |    92.13 (0.06)/ 88.41 (0.05)     |    91.39 (0.25)     |     90.27 (0.11)/ 90.3 (0.18)     |  **94.28 (0.11)**   | 87.36 (0.30) | 88.83|
> |    SoRA    |  0.91M  |  **71.48 (1.17)**   | 95.64 (0.23)|  **91.98 (1.16)**   |  **92.39 (0.17)/ 89.87 (0.27)**   |  **92.22  (0.24)**  | 90.35 (0.09)/ 90.38 (0.12)|  **94.28 (0.06)**   |  **87.77 (1.56)**   |  **89.36**   |
>
>
> - **Table 2**
>
> |   Method    | #Params |     CoLA     |    SST-2     |     MRPC     |            QQP             |     STS-B     |            MNLI            |     QNLI     |     RTE      | Avg.  |
> | :---------: | :-----: | :----------: | :----------: | :----------: | :------------------------: | :-----------: | :------------------------: | :----------: | :----------: | :---: |
> | LoRA (r=1)  |  0.17M  | 68.60 (1.58) | 94.95 (0.20) | 88.24 (1.42) | 91.20 (0.20)/ 88.37 (0.33) | 91.41 (0.29)  | 90.09 (0.16)/ 90.28 (0.14) | 93.35 (0.34) | 81.29 (2.83) | 87.23 |
> | SoRA (r=1)  |  0.12M  | 70.24 (1.10) | 95.14 (0.25) | 89.22 (2.12) | 91.52 (0.14)/ 88.73 (0.22) | 91.41 (0.29)  | 90.08 (0.18)/ 90.41 (0.12) | 93.43 (0.33) | 83.02 (2.47) | 87.85 |
> | LoRA (r=2)  |  0.33M  | 68.93 (1.50) | 95.04 (0.37) | 88.43 (1.06) | 91.59 (0.06)/ 88.87 (0.12) | 91.53 (0.32)  | 90.35 (0.13)/ 90.30 (0.17) | 93.63 (0.37) | 84.17 (1.20) | 87.79 |
> | SoRA (r=2)  |  0.25M  | 70.22 (1.03) | 95.64 (0.09) | 89.71 (0.81) | 91.88 (0.13)/ 89.12 (0.17) | 91.63 (0.30)  | 90.37 (0.06)/ 90.51 (0.08) | 93.78 (0.23) | 85.18 (0.73) | 88.39 |
> | LoRA (r=4)  |  0.66M  | 69.27 (2.22) | 95.55 (0.56) | 89.22 (1.64) | 91.40 (0.17)/ 88.41 (0.27) | 91.69 (0.24)  | 90.36 (0.12)/ 90.49 (0.32) | 93.83 (0.35) | 82.01 (1.76) | 87.74 |
> | SoRA (r=4)  |  0.47M  | 71.05 (0.74) | 95.57 (0.05) | 90.20 (1.60) | 92.06 (0.17)/ 89.44 (0.14) | 91.76 (0.17)  | 90.38 (0.04)/ 90.43 (0.25) | 93.92 (0.09) | 86.04 (1.96) | 88.71 |
> | LoRA (r=8)  |  1.33M  | 69.73 (1.42) | 95.57 (0.21) | 89.71 (1.32) | 91.95 (0.12)/ 89.26 (0.18) | 91.86 (0.29)  | 90.47 (0.23)/ 90.46 (0.12) | 93.76 (0.36) | 85.32 (0.86) | 88.38 |
> | SoRA (r=8)  |  0.91M  | 71.48 (1.17) | 95.64 (0.23) | 91.98 (1.16) | 92.39 (0.17)/ 89.87 (0.27) | 92.22  (0.24) | 90.35 (0.09)/ 90.38 (0.12) | 94.28 (0.06) | 87.77 (1.56) | 89.36 |
> | LoRA (r=16) |  2.65M  | 69.87 (0.86) | 95.53 (0.15) | 89.91 (1.69) | 92.22 (0.05)/ 89.63 (0.04) | 91.79 (0.16)  | 90.55 (0.10)/ 90.31 (0.03) | 93.46 (0.12) | 87.05 (3.11) | 88.62 |
> | SoRA (r=16) |  1.78M  | 71.93 (0.97) | 95.61 (0.11) | 92.00 (0.23) | 92.37 (0.15)/ 89.84 (0.19) | 92.05 (0.16)  | 90.34 (0.13)/ 90.47 (0.04) | 94.11 (0.07) | 87.41 (1.08) | 89.33 |
>
> - **Table 3**
>
> |                          | #Params |     CoLA     |     MRPC     |     STS-B     |
> | :----------------------- | :-----: | :----------: | :----------: | :-----------: |
> | $\mathbf{W}_{Q,K}$       |  0.80M  | 65.56 (2.24) | 86.43 (0.83) | 72.30 (2.61)  |
> | $\mathbf{W}_{Q,K,V}$     |  0.69M  | 69.07 (2.17) | 88.24 (1.84) | 90.82 (0.70)  |
> | $\mathbf{W}_{Q,K,V,A.O}$ |  0.87M  | 71.99 (1.30) | 90.20 (1.83) | 91.71 (0.34)  |
> | **All**                  |  0.77M  | 71.48 (1.17) | 91.98 (1.16) | 92.22  (0.24) |
>
> > How is $r_{max}$ defined? If it's set during initialization, shouldn't hyperparameter optimization be conducted?
> - Yes. $r_{max}$ is a hyperparameter standing for the maximum rank of SoRA. To compare SoRA experiments initialized with different  r_max, please find them in Table 2, in which the “$r$” of SoRA results is exactly the hyperparameter $r_{max}$. From Table 2, two conclusions can be drawn:
>     - For a given rank $r$, SoRA initialized with $r_{max}=r$ can steadily end up with better performance and fewer parameters than LoRA fixed with the same rank $r$.
>     - SoRA is highly robust to a wide range of $r_{max}$ choices. Slightly increasing $r_{max}$ generally introduces better results.
> In summary, we have demonstrated the tuning results of hyperparameter $r_{max}$ in Table 2, which consequently proves the robustness of SoRA to different  $r_{max}$ choices. Empirically, we suggest setting $r_{max}$ to be 1x $\sim$ 2x of the commonly used LoRA rank.
>
> > Does $\eta_t$ being larger than 0 or $\eta_t > 0$  represent the step size at the $t$-th iteration?
> - You are right. $\eta_t$ here is exactly the $t$-th step size of proximal gradient descent for the gate unit.
>
> > Your algorithm should also use \toprule, \midrule, and \bottomrule
> - Thank you very much for your suggestion. We have modified the manuscript and will update it in the next version of the paper.

---

### Official Review · Reviewer_rN3k · 2023-08-03

**Soundness:** 4

**Excitement:**

4: Strong: This paper deepens the understanding of some phenomenon or lowers the barriers to an existing research direction.

**Paper Topic And Main Contributions:**

The authors proposed a parameter-efficient method for large language model fine-tuning, SoRA (Sparse Low-Rank Adaptation), which aims to resolve the limitation of the existing method, Low-Rank Adaptation (LoRA (Hu et al., 2021)). This work's contribution includes introducing a flexible way to dynamically adjust the low intrinsic rank with theoretical support. Instead of viewing the rank of the lower-rank matrices $r$ as a hyperparameter, the authors add a gating unit $\textbf{g}$ between two trainable low-rank matrices (i.e., $B$ and $A$ in the original LoRA paper) with a fixed $r_{max}$. The introduced gating unit is updated by proximal gradient descent with a soft-thresholding function that sets values within $(-\xi,\xi]$ to zeros which can be dropped in the post-pruning phase, where $\xi$ is a changeable threshold for incrementally increasing model sparsity.
The results show that the proposed method SoRA can improve LoRA in 7 out of 8 tasks on the GLUE benchmark, retaining ~70% of parameters (Table 1) with the rank $r$ set to 8.

**Questions For The Authors:**

- Question A: Table 3 shows the `#Params` of \mathbf{W}_{Q,K} is 0.8M. However, the `#Params` of All, including the weights of $Q, K, V, A.O, f_1, f_2$ is 0.77M. Does `#Params` means the total parameters of the weight matrices after applying SoRA? If it is, why \mathbf{W}_{Q,K,V}'s 0.69M is smaller than \mathbf{W}_{Q,K}'s 0.80M? (It seems that the latex symbols do not render correctly, so I remove all latex $ here for better reading)
- Question B: Will the source code be public if accepted?


**Reasons To Accept:**

1. The proposed method SoRA enlights a way to apply the lower-rank adaptation method in practice. The rank analysis illustrated in Section 4.4 and Figure 3 shows that ranks vary across tasks and layers. Hence, compared to the identical rank in LoRA, SoRA has more potential to be extended to more applications.
2. Parameter-efficient fine-tuning of large language models has become critical as ChatGPT attracted wide attention recently. Incorporating large language models into applications extensively with various downstream tasks requires efficient fine-tuning as this work does.

**Reasons To Reject:**

1. The parameter $\xi$ for controlling model sparsity is worth exploring, as mentioned in the Limitations. More examples are required to understand the potential weakness of the proposed method. But overall, I do not have a strong reason to reject this paper.


**Reproducibility:**

5: Could easily reproduce the results.

**Reviewer Confidence:**

4: Quite sure. I tried to check the important points carefully. It's unlikely, though conceivable, that I missed something that should affect my ratings.

---

> ### Author Rebuttal · Authors · 2023-08-29
>
> We greatly appreciate your insightful comments and suggestions.
> > Question A: Table 3 shows the `#Params` of \mathbf{{W}_{Q,K}} is 0.8M. However, the #Params of All, including the weights of
>  is 0.77M. Does `#Params` means the total parameters of the weight matrices after applying SoRA? If it is, why \mathbf{W}{Q,K,V}'s 0.69M is smaller than \mathbf{W}_{Q,K}'s 0.80M? (It seems that the latex symbols do not render correctly, so I remove all latex $ here for better reading)
> - We sincerely apologize for not providing a detailed explanation of the specific meaning of `#Params` in the table caption. Throughout the paper, all instances of `#Params` related to SoRA refer to the number of non-zero parameters (parameters that will remain after training) in the Low-rank Adaptation after the Post-pruning, rather than the number of the initial parameter of the Low-rank Adaptation, corresponding to $W_d\in R^{r_{\text{max}} \times  q}$ and $W_u∈R^{p × r_{max}}$.
> This is because zero parameters will be removed after training in SoRA.
> - In the corresponding experiments of Table 3, to eliminate the influence of parameter quantity on results, it is necessary to ensure consistency in the number of non-zero parameters after the Post-pruning, when we apply SoRA to different weights. To achieve this, we introduce different values of $r_{max}$ in different experiments. For instance, when we apply SoRA to $W_{Q,K}$, $r_{max}$ is set to its maximum, while for applying SoRA to all weights, $r_{max}$ is minimized. In the experiments, we try our best to make the count of non-zero parameters (`#Params`) as consistent as possible, but ensuring uniformity in the `#Params` is extremely difficult, given that $r_{max}$ and the `#Params` do not exhibit a strict functional correspondence.
> - In a word, the `#Params` in Table 3 means the number of parameters that would remain after training, and it will be clearly stated in the next version of the paper. It should be kept as consistent as possible and has no relation to the weights that underwent the application of SoRA.
>
> > Will the source code be public if accepted?
> - Of course! We will include the GitHub repository link for the source code in the paper if accepted. We have proved in Table 2 that SoRA could be a consistently more effective and efficient method compared to the original LoRA. We would be pleased if you find our code useful and decide to utilize it.
>
> > The parameter for controlling model sparsity $\xi$ is worth exploring, as mentioned in the Limitations.
> - Yes, we feel it will be interesting to conduct relevant analysis (especially theoretical) about model sparsity in parameter-efficient fine-tuning. We will keep working on it. Thanks for you comments again.

---

### Official Review · Reviewer_kCU2 · 2023-08-09

**Soundness:** 4

**Excitement:**

4: Strong: This paper deepens the understanding of some phenomenon or lowers the barriers to an existing research direction.

**Missing References:**

[1]. [Enabling Lightweight Fine-tuning for Pre-trained Language Model Compression based on Matrix Product Operators](https://aclanthology.org/2021.acl-long.418) (Liu et al., ACL-IJCNLP 2021)

**Paper Topic And Main Contributions:**

This paper introduces a new approach called sparse low-rank adaptation (SoRA) that allows for dynamic adjustments to the intrinsic rank during the adaptation process, proposing a gate unit optimized with a proximal gradient method to control the sparsity of the adaptation, and demonstrating the effectiveness of SoRA on various natural language processing tasks.

**Questions For The Authors:**

1. Compare with Table 1, why are the results of QQP and MNLI missing in Table 2?
2. This paper does not provide a detailed analysis of the computational complexity of SoRA, which may limit its use in resource-constrained environments.

**Reasons To Accept:**

1. The proposed SoRA can dynamically adjust the intrinsic rank during the adaptation process, thereby improving parameter efficiency. This idea is quite novel.
2. SoRA introduces an optimizable gate unit that controls the sparsity of the adaptation using the proximal gradient method, thereby expanding the optimization space, this technique makes sense.
3. The authors demonstrate the effectiveness of SoRA on various natural language processing tasks, proving its superiority in terms of parameter efficiency and performance.
4. This paper is well written.

**Reasons To Reject:**

1. The comparison experiment with the MPOP method is lacking (See Missing References [1]). MPOP method is a lightweight fine-tuning method based on matrix decomposition and low-rank approximation, and it has a strong relevance to the method proposed in the paper. However, there is no comparison with this baseline method in the paper.
2. In Table 4, only the training time of the proposed method and AdaLoRA is compared, lacking the efficiency comparison with LoRA, Bitfit, and Adapter.
3. In the experimental section of the paper, the standard deviation after multiple experiments is not provided. The improvement brought by SoRA compared with the baseline is quite limited, which may be due to random fluctuations. The author should clarify which effects are within the range of standard deviation fluctuations and which are improvements brought by the SoRA method.

**Reproducibility:**

4: Could mostly reproduce the results, but there may be some variation because of sample variance or minor variations in their interpretation of the protocol or method.

**Reviewer Confidence:**

4: Quite sure. I tried to check the important points carefully. It's unlikely, though conceivable, that I missed something that should affect my ratings.

---

> ### Author Rebuttal · Authors · 2023-08-29
>
> Thanks for your insightful and constructive comments and suggestions.
>
> > The comparison experiment with the MPOP method is lacking (See Missing References [1]). MPOP method is a lightweight fine-tuning method based on matrix decomposition and low-rank approximation, and it has a strong relevance to the method proposed in the paper. However, there is no comparison with this baseline method in the paper.
> - We appreciate your advice. We have added the reference to MPOP in our revised paper. Yet we do not deem the relation between SoRA and MPOP a “strong relevance”. MPOP alleviates PLM computation cost by compressing the existing weights – a model compression approach; LoRA in contrast freezes those weights and tunes plugged-in modules on downstream tasks – a parameter-efficient fine-tuning (PEFT) methodology. SoRA, as an amelioration of LoRA, is consequently orthogonal to the realm of MPOP and other model compression methods. What is more, SoRA seeks for optimizable rank in the training process, as opposed to fixed-rank decomposition applied in either LoRA or MPOP. In spite of the lack of direct relevance, we acknowledge that MPOP makes use of the same mathematical truth – reduced-rank decomposition of matrices/tensors – as LoRA, and LoRA variants including our work SoRA. For this reason, we mention MPOP in our new version for a more comprehensive literature review. It will be updated in the revised version.
>
> > In Table 4, only the training time of the proposed method and AdaLoRA is compared, lacking the efficiency comparison with LoRA, Bitfit, and Adapter.
> - Thanks for your suggestion. We mainly compare SoRA to AdaLoRA, our primary baseline, to prove that our simpler method could achieve better performance and more efficiency when implementing adaptive rank. As mentioned in the last response (about complexity), the time complexity is approximately the same as LoRA with slightly more overheads. In fact, the gradient computation of SoRA can further benefit from the algorithmic optimization of deep learning frameworks since many of the matrices involved are either row-sparse or column-sparse due to the stop-gradient effect of gating. We provide the training time for all five methods in the table below. Detailed analysis of the training time for SoRA and LoRA can be found in the answer to the fifth question.
>
> | Datasets | AdaLoRA (second) | SoRA (second) | LoRA (second) | Adapter (second) | Bitfit (second) |
> | :------: | :--------------: | :-----------: | :-----------: | :--------------: | :-------------: |
> |   CoLA   |      160.2       |     57.2      |     51.0      |       53.3       |      38.4       |
> |  SST-2   |      491.0       |     433.0     |     379.0     |      413.4       |      297.7      |
> |   MRPC   |       27.3       |     24.8      |     23.2      |       23.6       |      16.5       |
> |  STS-B   |       48.2       |     38.4      |     33.3      |       34.7       |      25.9       |
> |   QNLI   |      1001.0      |     676.3     |     594.9     |      655.0       |      478.9      |
> |   RTE    |       79.8       |     45.1      |     17.9      |       15.3       |      11.3       |
> |   Avg.   |      301.3       |     212.5     |     183.2     |      199.2       |      144.8      |
>
> > In the experimental section of the paper, the standard deviation after multiple experiments is not provided. The improvement brought by SoRA compared with the baseline is quite limited, which may be due to random fluctuations. The author should clarify which effects are within the range of standard deviation fluctuations and which are improvements brought by the SoRA method.
> - Thank you for the reminder. We have supplemented the standard deviation in the table below, which will be included in the next version of the paper. The standard deviation corresponding to SoRA is not significantly different from other methods, indicating that the improvement in performance on the GLUE tasks is indeed attributed to the SoRA method.
>
> - **Table 1**
>
> |   Method   | #Params |        CoLA         |        SST-2        |        MRPC         |                QQP                |        STS-B        |               MNLI                |        QNLI         |         RTE         |     Avg.     |
> | :--------: | :-----: | :-----------------: | :-----------------: | :-----------------: | :-------------------------------: | :-----------------: | :-------------------------------: | :-----------------: | :-----------------: | :----------: |
> | Fine-Tune  |  184M   |    69.21 (2.24)     |    95.64 (0.52)     |    89.22 (0.69)     | 92.05 (0.09)/ 89.31 (0.07)|    91.59 (0.47)     |    89.98 (0.06)/ 89.95 (0.33)     |    93.78 (0.02)     |    82.49 (1.48)     |    87.82     |
> |  Adapter   |  1.41M  |    69.00 (0.91)     |    95.16 (0.46)     |    89.90 (2.10)     |    91.45 (0.18)/ 88.88 (0.40)     | 92.21 (0.33)|    90.11 (0.57)/ 90.11 (0.57)     | 93.79 (0.07)|    82.44 (1.74)     |    87.85     |
> |   Bitfit   |  0.1M   |    68.70 (1.85)     |    94.38 (0.28)     |    87.16 (0.58)     |    87.86 (0.52)/ 84.20 (0.74)     |    89.71 (0.58)     |    87.45 (0.76)/ 87.45 (0.76)     |    91.90 (0.14)     |    76.12 (1.54)     |    85.18     |
> | LoRA (r=8) |  1.33M  |    69.73 (1.42)     |    95.57 (0.21)     |    89.71 (1.32)     |    91.95 (0.12)/ 89.26 (0.18)     |    91.86 (0.29)     |  **90.47 (0.23)/ 90.46 (0.12 )**  |    93.76 (0.36)     |    85.32 (0.86)     |    88.38     |
> |  AdaLoRA   |  1.27M  | 70.86 (1.43)|  **95.95 (0.37)**   | 90.22 (0.40) |    92.13 (0.06)/ 88.41 (0.05)     |    91.39 (0.25)     |     90.27 (0.11)/ 90.3 (0.18)     |  **94.28 (0.11)**   | 87.36 (0.30) | 88.83|
> |    SoRA    |  0.91M  |  **71.48 (1.17)**   | 95.64 (0.23)|  **91.98 (1.16)**   |  **92.39 (0.17)/ 89.87 (0.27)**   |  **92.22  (0.24)**  | 90.35 (0.09)/ 90.38 (0.12)|  **94.28 (0.06)**   |  **87.77 (1.56)**   |  **89.36**   |
>
> - **Table 2**
>
> |   Method    | #Params |     CoLA     |    SST-2     |     MRPC     |            QQP             |     STS-B     |            MNLI            |     QNLI     |     RTE      | Avg.  |
> | :---------: | :-----: | :----------: | :----------: | :----------: | :------------------------: | :-----------: | :------------------------: | :----------: | :----------: | :---: |
> | LoRA (r=1)  |  0.17M  | 68.60 (1.58) | 94.95 (0.20) | 88.24 (1.42) | 91.20 (0.20)/ 88.37 (0.33) | 91.41 (0.29)  | 90.09 (0.16)/ 90.28 (0.14) | 93.35 (0.34) | 81.29 (2.83) | 87.23 |
> | SoRA (r=1)  |  0.12M  | 70.24 (1.10) | 95.14 (0.25) | 89.22 (2.12) | 91.52 (0.14)/ 88.73 (0.22) | 91.41 (0.29)  | 90.08 (0.18)/ 90.41 (0.12) | 93.43 (0.33) | 83.02 (2.47) | 87.85 |
> | LoRA (r=2)  |  0.33M  | 68.93 (1.50) | 95.04 (0.37) | 88.43 (1.06) | 91.59 (0.06)/ 88.87 (0.12) | 91.53 (0.32)  | 90.35 (0.13)/ 90.30 (0.17) | 93.63 (0.37) | 84.17 (1.20) | 87.79 |
> | SoRA (r=2)  |  0.25M  | 70.22 (1.03) | 95.64 (0.09) | 89.71 (0.81) | 91.88 (0.13)/ 89.12 (0.17) | 91.63 (0.30)  | 90.37 (0.06)/ 90.51 (0.08) | 93.78 (0.23) | 85.18 (0.73) | 88.39 |
> | LoRA (r=4)  |  0.66M  | 69.27 (2.22) | 95.55 (0.56) | 89.22 (1.64) | 91.40 (0.17)/ 88.41 (0.27) | 91.69 (0.24)  | 90.36 (0.12)/ 90.49 (0.32) | 93.83 (0.35) | 82.01 (1.76) | 87.74 |
> | SoRA (r=4)  |  0.47M  | 71.05 (0.74) | 95.57 (0.05) | 90.20 (1.60) | 92.06 (0.17)/ 89.44 (0.14) | 91.76 (0.17)  | 90.38 (0.04)/ 90.43 (0.25) | 93.92 (0.09) | 86.04 (1.96) | 88.71 |
> | LoRA (r=8)  |  1.33M  | 69.73 (1.42) | 95.57 (0.21) | 89.71 (1.32) | 91.95 (0.12)/ 89.26 (0.18) | 91.86 (0.29)  | 90.47 (0.23)/ 90.46 (0.12) | 93.76 (0.36) | 85.32 (0.86) | 88.38 |
> | SoRA (r=8)  |  0.91M  | 71.48 (1.17) | 95.64 (0.23) | 91.98 (1.16) | 92.39 (0.17)/ 89.87 (0.27) | 92.22  (0.24) | 90.35 (0.09)/ 90.38 (0.12) | 94.28 (0.06) | 87.77 (1.56) | 89.36 |
> | LoRA (r=16) |  2.65M  | 69.87 (0.86) | 95.53 (0.15) | 89.91 (1.69) | 92.22 (0.05)/ 89.63 (0.04) | 91.79 (0.16)  | 90.55 (0.10)/ 90.31 (0.03) | 93.46 (0.12) | 87.05 (3.11) | 88.62 |
> | SoRA (r=16) |  1.78M  | 71.93 (0.97) | 95.61 (0.11) | 92.00 (0.23) | 92.37 (0.15)/ 89.84 (0.19) | 92.05 (0.16)  | 90.34 (0.13)/ 90.47 (0.04) | 94.11 (0.07) | 87.41 (1.08) | 89.33 |
>
> - **Table 3**
>
> |                          | #Params |     CoLA     |     MRPC     |     STS-B     |
> | :----------------------- | :-----: | :----------: | :----------: | :-----------: |
> | $\mathbf{W}_{Q,K}$       |  0.80M  | 65.56 (2.24) | 86.43 (0.83) | 72.30 (2.61)  |
> | $\mathbf{W}_{Q,K,V}$     |  0.69M  | 69.07 (2.17) | 88.24 (1.84) | 90.82 (0.70)  |
> | $\mathbf{W}_{Q,K,V,A.O}$ |  0.87M  | 71.99 (1.30) | 90.20 (1.83) | 91.71 (0.34)  |
> | **All**                  |  0.77M  | 71.48 (1.17) | 91.98 (1.16) | 92.22  (0.24) |
>
> - The results of the standard deviation indicate that the improvement is not brought by fluctuations. Also, we think the improvement in GLUE tasks at this range could be considered non-trivial.
>
> > Compared with Table 1, why are the results of QQP and MNLI missing in Table 2?
> - QQP and MNLI tasks are very resource-intensive and we feel the results of the other six datasets could sufficiently demonstrate effectiveness at the time of preparing the manuscript for analysis. However, we have supplemented the results in the table below. Thanks for your suggestion!
>
> - **Table 2**
> |   Method    | #Params |     CoLA     |    SST-2     |     MRPC     |            QQP             |     STS-B     |            MNLI            |     QNLI     |     RTE      | Avg.  |
> | :---------: | :-----: | :----------: | :----------: | :----------: | :------------------------: | :-----------: | :------------------------: | :----------: | :----------: | :---: |
> | LoRA (r=1)  |  0.17M  | 68.60 (1.58) | 94.95 (0.20) | 88.24 (1.42) | 91.20 (0.20)/ 88.37 (0.33) | 91.41 (0.29)  | 90.09 (0.16)/ 90.28 (0.14) | 93.35 (0.34) | 81.29 (2.83) | 87.23 |
> | SoRA (r=1)  |  0.12M  | 70.24 (1.10) | 95.14 (0.25) | 89.22 (2.12) | 91.52 (0.14)/ 88.73 (0.22) | 91.41 (0.29)  | 90.08 (0.18)/ 90.41 (0.12) | 93.43 (0.33) | 83.02 (2.47) | 87.85 |
> | LoRA (r=2)  |  0.33M  | 68.93 (1.50) | 95.04 (0.37) | 88.43 (1.06) | 91.59 (0.06)/ 88.87 (0.12) | 91.53 (0.32)  | 90.35 (0.13)/ 90.30 (0.17) | 93.63 (0.37) | 84.17 (1.20) | 87.79 |
> | SoRA (r=2)  |  0.25M  | 70.22 (1.03) | 95.64 (0.09) | 89.71 (0.81) | 91.88 (0.13)/ 89.12 (0.17) | 91.63 (0.30)  | 90.37 (0.06)/ 90.51 (0.08) | 93.78 (0.23) | 85.18 (0.73) | 88.39 |
> | LoRA (r=4)  |  0.66M  | 69.27 (2.22) | 95.55 (0.56) | 89.22 (1.64) | 91.40 (0.17)/ 88.41 (0.27) | 91.69 (0.24)  | 90.36 (0.12)/ 90.49 (0.32) | 93.83 (0.35) | 82.01 (1.76) | 87.74 |
> | SoRA (r=4)  |  0.47M  | 71.05 (0.74) | 95.57 (0.05) | 90.20 (1.60) | 92.06 (0.17)/ 89.44 (0.14) | 91.76 (0.17)  | 90.38 (0.04)/ 90.43 (0.25) | 93.92 (0.09) | 86.04 (1.96) | 88.71 |
> | LoRA (r=8)  |  1.33M  | 69.73 (1.42) | 95.57 (0.21) | 89.71 (1.32) | 91.95 (0.12)/ 89.26 (0.18) | 91.86 (0.29)  | 90.47 (0.23)/ 90.46 (0.12) | 93.76 (0.36) | 85.32 (0.86) | 88.38 |
> | SoRA (r=8)  |  0.91M  | 71.48 (1.17) | 95.64 (0.23) | 91.98 (1.16) | 92.39 (0.17)/ 89.87 (0.27) | 92.22  (0.24) | 90.35 (0.09)/ 90.38 (0.12) | 94.28 (0.06) | 87.77 (1.56) | 89.36 |
> | LoRA (r=16) |  2.65M  | 69.87 (0.86) | 95.53 (0.15) | 89.91 (1.69) | 92.22 (0.05)/ 89.63 (0.04) | 91.79 (0.16)  | 90.55 (0.10)/ 90.31 (0.03) | 93.46 (0.12) | 87.05 (3.11) | 88.62 |
> | SoRA (r=16) |  1.78M  | 71.93 (0.97) | 95.61 (0.11) | 92.00 (0.23) | 92.37 (0.15)/ 89.84 (0.19) | 92.05 (0.16)  | 90.34 (0.13)/ 90.47 (0.04) | 94.11 (0.07) | 87.41 (1.08) | 89.33 |
>
> > This paper does not provide a detailed analysis of the computational complexity of SoRA, which may limit its use in resource-constrained environments.
> - Thanks for the suggestion. Compared to LoRA, SoRA introduces additional computation only in the gate control part, and only during training, not inference. In forward propagation, the gating operation is simply a vector-by-vector element-wise multiplication, which takes merely $\mathcal{O}(r_{\text{max}})$ time. As for backward propagation, the change in gradient computation complexity is difficult to explicitly formulate on PLM. But our experiments show that SoRA merely requires 10% more wall-clock training time than LoRA. In fact, the gradient computation of SoRA can further benefit from the algorithmic optimization of deep learning frameworks since many of the matrices involved are either row-sparse or column-sparse due to the stop-gradient effect of gating. Similarly, the increase in training-phase storage will be in the same order $\mathcal{O}(r_{\text{max}})$ since the optimizable gate is the only weight additional to LoRA. Our experimental results can also help demonstrate that SoRA is approximately as resource-economical as LoRA.

---

### Meta-Review · Area_Chair_hrSz · 2023-09-08

**Recommendation:** 5

**Metareview:**

There is strong consensus that this work is both very exciting and very sound. The reviewers agree that the idea and its intuition is important and insightful and that the experimental results are excellent. As such, I recommend acceptance to the main conference.

---

### Decision · Program_Chairs · 2023-10-07

**Decision:**

Accept-Main

**Comment:**

There is strong consensus that this work is both very exciting and very sound. The reviewers agree that the idea and its intuition is important and insightful and that the experimental results are excellent. As such, I recommend acceptance to the main conference.